# Clinicopathological Significance and Prognostic Implications of REG4 Immunohistochemical Expression in Colorectal Cancer

**DOI:** 10.3390/medicina57090938

**Published:** 2021-09-05

**Authors:** Guhyun Kang, Ilhwan Oh, Jungsoo Pyo, Dongwook Kang, Byoungkwan Son

**Affiliations:** 1Department of Pathology, Daehang Hospital, Seoul 06699, Korea; guhyunkang@daum.net; 2Department of Internal Medicine, Uijeongbu Eulji Medical Center, Eulji University School of Medicine, Uijeongbu-si 11759, Korea; 20180121@eulji.ac.kr; 3Department of Pathology, Uijeongbu Eulji Medical Center, Eulji University School of Medicine, Uijeongbu-si 11759, Korea; jspyo@eulji.ac.kr; 4Department of Pathology, Chungnam National University Sejong Hospital, 20 Bodeum 7-ro, Sejong 30099, Korea; astro966@gmail.com; 5Department of Pathology, Chungnam National University School of Medicine, 266 Munhwa Street, Daejeon 35015, Korea

**Keywords:** colorectal cancer, REG4, immunohistochemistry, tumor stroma, prognosis

## Abstract

*Background and objectives:* The present study aimed to evaluate the clinicopathological significance and prognostic implications of REG4 immunohistochemical expression in colorectal cancer (CRC). *Materials and Methods:* We performed immunohistochemical analysis for REG4 cytoplasmic expression in 266 human CRC tissues. Correlations between REG4 expression, clinicopathological characteristics, and survival were investigated in CRC. *Results:* REG4 was expressed in 84 of 266 CRC tissues (31.6%). REG4 expression was significantly more frequent in the right colon than that in the left colon and rectum (*p* = 0.002). However, we observed no significant correlation between REG4 expression and other clinicopathological parameters. REG4 expression was significantly higher in CRCs with low stroma than in those with high stroma (*p* = 0.006). In addition, REG4 was more frequently expressed in CRCs with the mucinous component than in those without it (*p* < 0.001). There was no significant correlation between REG4 expression and overall recurrence-free survival (*p* = 0.132 and *p* = 0.480, respectively). Patients with REG4 expression showed worse overall and recurrence-free survival in the high-stroma subgroup (*p* = 0.001 and *p* = 0.017, respectively), but no such correlation was seen in the low stroma subgroup (*p* = 0.232 and *p* = 0.575, respectively). *Conclusions:* REG4 expression was significantly correlated with tumor location, amount of stroma, and mucinous component in CRCs. In patients with high stroma, REG4 expression was significantly correlated with poor overall and recurrence-free survival.

## 1. Introduction

Regenerating islet-derived (REG) proteins, a group of small secretory proteins, contain a carbohydrate-recognition domain [1]. The REG family includes proteins from REG1 to REG4 [2]. REG proteins are involved in cell regeneration and proliferation [3,4]. Among the REG proteins, REG4, which is the most recently discovered member, is a small, 17 kD secreted C-type lectin [5]. In malignant tumor cells, REG4 promotes proliferation and inhibits apoptosis [6,7,8]. Previous studies have reported the prognostic or predictive role of REG4 in gastrointestinal malignancies [9,10,11,12,13]. REG4 is expressed in normal enteric neuroendocrine cells as well as in some goblet cells of the stomach, small intestine, and colon [14,15]. In the gallbladder, REG4 is expressed in the intestinal metaplastic epithelium and cancer cells [16]. REG4 is expressed in various diseases or tumors, such as inflammatory bowel diseases [17] and gastrointestinal malignancies [10,11,15,18,19,20,21,22]. In addition, REG4 is also expressed in colonic adenomas [21]. Gao et al. showed the predictive role of REG4 in radiochemotherapy sensitivity in colorectal cancers (CRCs). However, the clinicopathological and prognostic implications of tissue REG4 expression in CRC remain controversial. CRC is a heterogeneous disease, which involves a multi-step process [23,24]. In addition, several genetic mutations can induce the malignant transformation of normal mucosa into cancer [23,24].

This study aimed to evaluate the clinicopathological significance of REG4 expression in colorectal cancer (CRC) using immunohistochemistry. Moreover, the prognostic implications of REG4 expression were investigated in CRC. The correlations between REG4 expression and amount of tumor stroma as well as the mucinous component were evaluated. In addition, to elucidate the correlation between REG4 and stroma of tumor, the implication of HIF-1α expression was investigated. Based on tumor stroma and mucinous component subgroups, prognostic stratification of CRC with and without REG4 expression was attempted.

## 2. Materials and Methods

### 2.1. Patients

Between 1 January 2001 and 31 December 2010, 266 patients who had undergone surgical resection for CRC at the Eulji University Medical Center were enrolled in this study. We reviewed the medical charts, pathological records, and glass slides to assess the following clinicopathological characteristics: age, sex, tumor size, tumor location, tumor differentiation, vascular, lymphatic and perineural invasion, depth of tumor, lymph node metastasis, metastatic lymph node ratio, distant metastasis, and pathologic tumor node metastasis (pTNM) stage. We evaluated these cases according to the 8th edition of the American Joint Cancer Committee TNM classification [25]. This protocol was reviewed and approved by the Institutional Review Board of Eulji University Hospital (Approval No. EMCIRB 2020-01-009). Clinical outcomes were followed from the date of surgery to either the date of death or recurrence, resulting in a follow-up period ranging from 0 to 60 months.

### 2.2. Tissue Array Method

Five array blocks containing a total of 266 resected colorectal cancer tissue cores obtained from patients were prepared. Briefly, we collected tissue cores (2 mm in diameter) from individual paraffin-embedded CRC tissues (donor blocks) using a trephine and arranged them in recipient paraffin blocks, as previously described [26]. The staining results for various intratumoral areas in these tissue-array blocks were highly consistent. A core was chosen from each case for the analysis. An adequate case was defined as a tumor occupying more than 10% of the core area. Each block contained internal controls consisting of non-neoplastic colon tissue.

### 2.3. Immunohistochemical Staining and Evaluation

Sections (4 μm thick) for immunohistochemical analysis were cut from each tissue array block, deparaffinized, and dehydrated. Immunohistochemical staining was performed following a compact polymer method using VENTANA BenchMark XT autostainer (Ventana Medical Systems, Inc., Tucson, AZ, USA). We then incubated the sections with anti-REG4 (Abcam, Cambridge, UK) and anti-HIF-1α (Novus Biologicals, Littleton, CO, USA) antibodies. Visualization was performed by treating the samples using an OptiView universal 3,3′-diaminobenzidine kit (Ventana Medical Systems, Inc., Tucson, AZ, USA). We used a negative control stain without a primary antibody to confirm the specificity of each antibody. All immuno-stained sections were lightly counterstained with Mayer’s hematoxylin. Immunohistochemical staining for REG4 and HIF-1α was detected in the cytoplasm and nucleus, respectively. The intensity of protein expression in the immunohistochemically stained samples was scored from 0 to 3 (0 = negative; 1 = weak; 2 = moderate; and 3 = strong). The percentage of positively stained cells was categorized using a scoring system with points ranging from 0 to 4 (1 = 0–25%; 2 = 26–50%; 3 = 51–75%; and 4 = 76–100%). The immunoreactive score (IRS) was calculated by multiplying the staining intensity scores by the percentage of positively stained cells [27]. The staining patterns of REG4 were classified as either negative (IRS: 0–2) or positive (IRS: 3–12). For the interpretation of HIF-1α expression, staining patterns were classified as either negative (IRS: 0–4) or positive (IRS: 6–12). The immunohistochemistry results were evaluated by two independent pathologists (G.Kang and J.Pyo). Cases with inconsistent results were reviewed and confirmed by agreement between two pathologists. However, for the qualitative categorization of the samples, the interpretation of blinded fashion could not be performed.

### 2.4. Evaluation of Intratumoral Stroma

The intratumoral stroma was evaluated using prepared glass slides with hematoxylin and eosin (H&E) staining. Because the amount of stroma can be different within the tumor, all H&E slides were screened and evaluated for the intratumoral stroma. The assessment of stroma was conducted using scoring percentages in 10% increments (10%, 20%, 30%, etc.). We divided the intratumoral stroma into high and low subgroups using a cutoff of 50%. Conflicting cases were re-examined, and consensus between three pathologists (Kang G., Pyo J., and Kang D.) was reached.

### 2.5. Statistical Analysis

Statistical analyses were performed using SPSS version 22.0 software (IBM Co., Chicago, IL, USA). The significance of the correlation between REG4 expression and clinicopathological characteristics was determined using the Chi-squared (χ2) test (two-sided). The comparisons between REG4 expression and age, tumor size, or metastatic lymph node ratio were analyzed using two-tailed Student’s *t*-test. In addition, the correlations between REG4 and intratumoral stroma and mucinous component were evaluated using the Chi-squared (χ2) test (two-sided). Survival curves were estimated using the Kaplan–Meier product-limit method, and differences between the survival curves were determined to be significant based on the log-rank test. Results were considered statistically significant at *p* < 0.05.

## 3. Results

### 3.1. Correlation between REG4 Expression and Clinicopathological Characteristics in Colorectal Cancer

Representative images of REG4 expression in human CRC tissues are shown in Figure 1. REG4 expression was significantly higher in the cancers of right colon than that in cancers of left colon and rectum (*p* = 0.002; Table 1). However, there was no significant correlation with other clinicopathological characteristics. REG4 expression was significantly higher in patients with low stroma than that in those with high stroma (*p* = 0.006; Table 2). REG4 was observed to have been expressed in 68 of 185 cases with low stroma and in 16 of 81 cases with high stroma. REG4 expression was significantly higher in the subgroup with the mucinous component than in the subgroup without it (*p* < 0.001). In addition, there was a significant correlation between REG4 expression and HIF-1α expression (*p* = 0.011).

### 3.2. Correlation between REG4 Expression and Prognosis

The prognostic implications of REG4 expression were evaluated in patients with CRC. There was no significant correlation between REG4 expression and overall and recurrence-free survival (*p* = 0.132 and *p* = 0.480, respectively; Figure 2). Next, an additional analysis based on the stroma subgroup was performed. In the high stroma subgroup, patients with REG4 expression had worse overall and recurrence-free survival than those without REG4 expression (*p* = 0.001 and *p* = 0.017, respectively; Figure 3). In contrast, there was no significant difference in survival between patients with and without REG4 expression in CRCs with low stroma (*p* = 0.232 and *p* = 0.575, respectively). In addition, there was no significant correlation between REG4 expression and prognosis according to subgrouping based on the mucinous component.

## 4. Discussion

In previous studies, the expression of REG proteins was correlated with tissue inflammation and carcinogenesis [1,28]. In the gastrointestinal tract, REG4 expression was found to have increased in inflammatory bowel diseases (IBD) [5,14], as well as in colorectal [12], gastric [10], and pancreatic cancer [29]. In addition, compared with normal colorectal tissues, REG4 gene expression was observed to have increased in colorectal cancer tissues [30]. However, the clinicopathological significance of REG4 protein expression in CRC tissues remains controversial. The present study aimed to elucidate the clinicopathological and prognostic implications of REG4 expression through immunohistochemical analysis of human CRC tissues. In addition, the correlation between REG4 expression and tumor stroma was investigated.

REG family members are thought to be involved in the proliferation and differentiation of various cell types, such as gastric, intestinal, hepatic, and pancreatic cells [31]. In tumor cells, REG4 expression is associated with cell growth, survival, and anti-apoptosis [6,11,32,33,34,35]. In gastric cancer-initiating cells, upregulation of REG4 has been identified [31]. Kawasaki et al. reported that REG4 could be involved in tumorigenesis through the mediation of GATA6 [36]. In addition, Zhang et al. suggested that REG4 expression may be associated with early CRC carcinogenesis [22]. If REG4 is associated with carcinogenesis, evaluation of its diagnostic role can be of importance in early CRC. Oue et al. studied the diagnostic role of serum REG4 levels in CRC. They reported that REG4 levels were not significantly increased in patients with stage III CRC [15]. REG4 mRNA levels were found to be significantly higher in adenoma than in the normal mucosa of the colon. In addition, the study also reported that there was no significant difference in REG4 mRNA levels between colorectal adenoma and adenocarcinoma [22]. In other words, the diagnostic role of REG4 in CRC remains unclear based on the previous studies. 

Among the 266 cases in our study, REG4 was expressed in 31.6% of CRC cases. Oue et al. reported that REG4 expression was found in 28.8% of 80 CRCs [15]. REG4 expression was significantly correlated with well-differentiated histological types and liver metastasis in CRC [15]. Numata et al. reported that high REG4 gene expression was significantly correlated with histologic differentiation, depth of invasion, lymphatic invasion, tumor stage, and liver metastasis [37]. However, there was no significant correlation between REG4 and tumor size, tumor location, lymph node metastasis, or venous invasion [37]. If REG4 is indeed important in tumorigenesis of CRC, the clinicopathological characteristics of CRCs with and without REG4 expression cannot be significant. In our study, REG4 expression was significantly correlated with tumor location. No significant correlation was observed between REG4 expression and other clinicopathological parameters.

In the present study, there was no significant correlation between REG4 expression and overall and recurrence-free survival (*p* = 0.132 and *p* = 0.480, respectively). Because various factors can affect the prognosis of CRC patients, detailed analyses of the prognostic role of REG4 expression are needed. We previously reported the clinicopathological implications of tumor stroma in CRC [38]. In the present study, CRCs with low stroma showed significantly higher REG4 expression than those with high stroma. In addition, in the high stroma subgroup, REG4 expression was significantly correlated with poor survival. However, there was no significant correlation between REG4 expression and prognosis in CRCs with low stroma. Although CRC with high stroma was significantly correlated with low REG4 expression, patients with CRCs with high stroma and REG4 expression showed worse survival than other CRC patients. REG4 expression could be stratified according to the prognosis of CRCs with high stroma. The prognostic implication of REG4 expression can differ according to the amount of intratumoral stroma. Therefore, in a study regarding the clinicopathological significance of REG4 expression, it is important to evaluate the amount of stroma. Thus, the interaction between REG4 expression and the stroma may be important. We also studied the correlation between REG4 and HIF-1α expression in CRC. HIF-1α expression was significantly higher in CRCs with high stroma and REG4 expression than that in the other subgroups. However, in the present study, there were no significant differences in survival between patients with CRCs with and without HIF-1α expression. The correlation between REG4 and HIF-1α expression has not been previously reported. Although our study could not completely elucidate the detailed pathway, it is the first to demonstrate the correlation between REG4 and HIF-1α expression through immunohistochemical analysis. Moreover, because REG4 can be correlated with inflammation, the correlation between REG4 and tumor-infiltrating lymphocytes was investigated. REG4 expression was not significantly correlated with tumor-infiltrating lymphocytes (*p* = 0.790). After the malignant transformation, the impact of REG4 on the inflammatory process can be limited. In addition, REG4 expression was not correlated with PD-L1 expressions of tumor and immune cells (*p* = 0.259 and *p* = 0.227, respectively).

Kaprio et al. reported that REG4 expression was associated with favorable clinicopathological parameters and prognosis in non-mucinous CRC [39]. In addition, REG4 expression was found to be correlated with intestinal mucinous markers, including MUC1, MUC2, and MUC5AC [39]. A previous study reported high serum REG4 levels in mucinous ovarian cancer [40]. Interestingly, in the present study, REG4 expression was significantly higher in CRCs with the mucinous component than in those without it. However, in CRCs with mucinous component, there was no significant difference in overall and recurrence-free survival between subgroups with or without REG4 expression (*p* = 0.447 and *p* = 0.894, respectively). Kaprio et al. suggested that REG4 is useful for predicting a better prognosis in non-mucinous CRC [39]. However, in the present study, we observed no significant difference in overall and recurrence-free survival between non-mucinous CRCs with and without REG4 expression (*p* = 0.079 and *p* = 0.284, respectively). 

The clinicopathological implications of REG4 expression have been studied in various malignant tumors. REG4 has been studied by dividing tissue expression and serum levels. In pancreatic ductal adenocarcinomas, tissue expression of REG4 was found to be a useful prognostic marker [2]. REG4 expression, which is increased in pancreatic cancer cells compared with normal pancreatic cells, can promote the invasiveness and proliferation of tumor cells [29,40,41]. In addition, serum REG4 levels may be useful for differentiating between pancreatic ductal adenocarcinoma and chronic pancreatitis [2]. In gastric cancers, serum REG4 levels were found to be significantly higher than those in healthy individuals [11,18]. Oue et al. reported that REG4-positive CRCs had a worse prognosis than REG4-negative CRCs. However, because this study included only 30 patients with CRC, conclusive data could not be obtained [12]. In the present study, we observed no significant correlation between REG4 expression and distant metastasis (*p* = 0.947). The diagnostic and prognostic roles of REG4 expression can differ according to the tumor type and researchers.

## 5. Conclusions

In conclusion, REG4 was found to be frequently expressed in right-sided CRC. In addition, REG4 expression was significantly higher in CRCs with low stroma than that in those with high stroma. According to the amount of stroma, REG4 expression was significantly correlated with poor overall and recurrence-free survival.

## Figures and Tables

**Figure 1 medicina-57-00938-f001:**
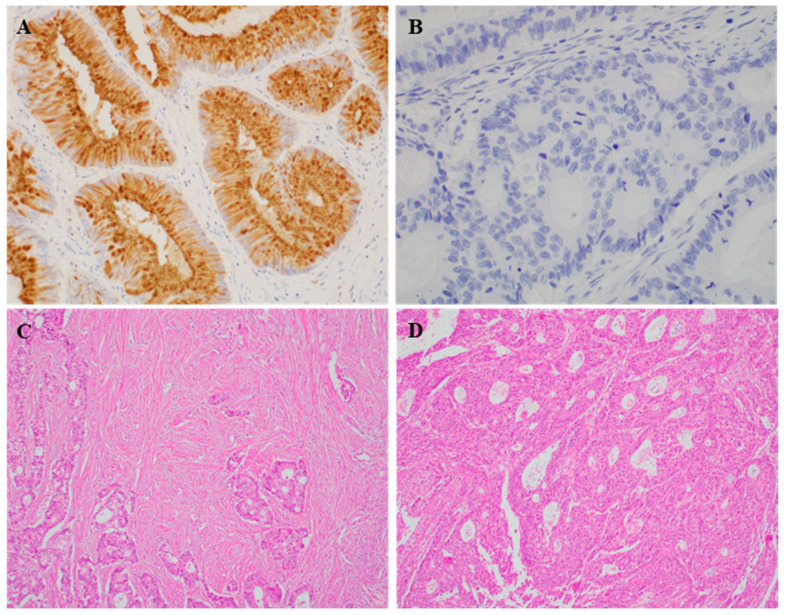
Representative immunohistochemical staining images for REG4 and intratumoral stroma. (**A**,**B**) Positive and negative REG4 expression in tumor cells (×400). (**C**,**D**) Low and high intratumoral stroma in colorectal cancer cases (×200).

**Figure 2 medicina-57-00938-f002:**
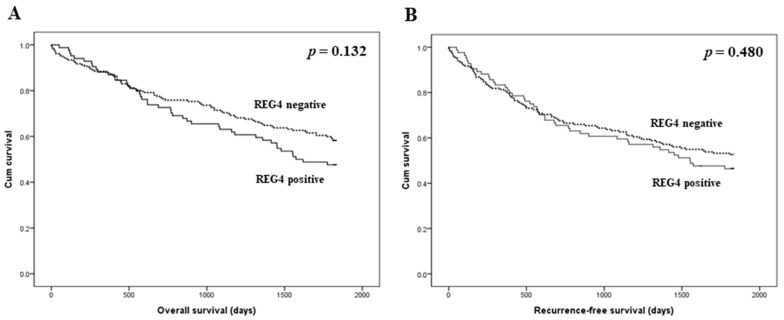
Kaplan-Meier analysis of patient survival according to REG4 expression in overall cases. (**A**) Overall survival. (**B**) Recurrence-free survival.

**Figure 3 medicina-57-00938-f003:**
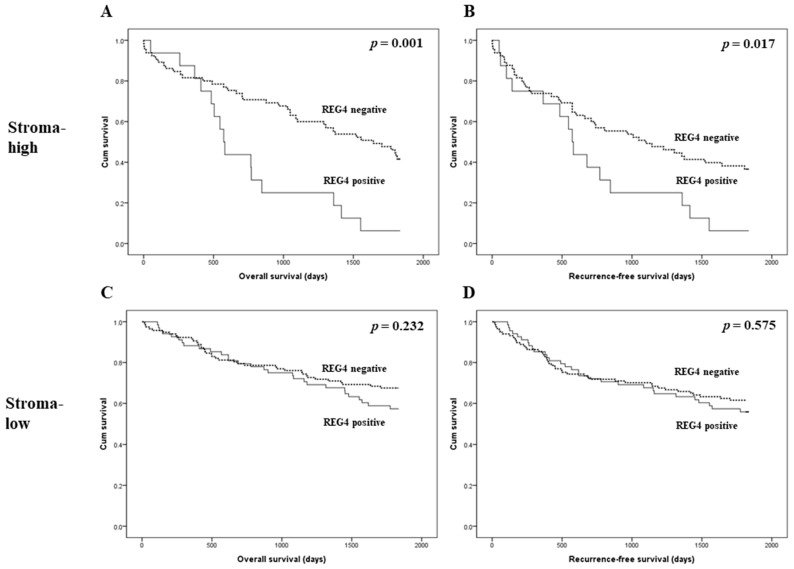
Kaplan-Meier analysis of patient survival according to REG4 expression and intratumoral stroma in colorectal cancer. (**A**,**B**) Overall and recurrence-free survival according to REG4 expression in colorectal cancer cases with high stroma. (**C**,**D**) Overall and recurrence-free survival according to REG4 expression in colorectal cancer cases with low stroma.

**Table 1 medicina-57-00938-t001:** The correlation between REG4 expression and clinicopathological parameters in colorectal cancers.

	REG4 Expression		*p*-Value
	Positive	Negative	
Total (*n* = 266)	84 (31.6)	182 (68.4)	
Age (years)	64.06 ± 12.89	63.37 ± 12.95	0.686
Sex Male Female	42 (50.0) 42 (50.0)	93 (51.1) 89 (48.9)	0.868
Tumor size ≤5 cm >5 cm	32 (38.1) 52 (61.9)	74 (40.7) 108 (59.3)	0.691
Tumor size (cm)	5.80 ± 2.16	5.30 ± 2.02	0.069
Location of tumor Right colon Left colon and rectum	52 (61.9) 32 (38.1)	76 (41.8) 106 (58.2)	0.002
Tumor differentiation Well or Moderate Poorly	62 (73.8) 22 (26.2)	149 (81.9) 33 (18.1)	0.131
Vascular invasion Present Absent	6 (7.1) 78 (92.9)	18 (9.9) 164 (90.1)	0.467
Lymphatic invasion Present Absent	16 (19.0) 68 (81.0)	54 (29.7) 128 (70.3)	0.067
Perineural invasion Present Absent	9 (10.7) 75 (89.3)	35 (19.2) 147 (80.8)	0.082
pT stage pT1-2 pT3-4	18 (21.4) 66 (78.6)	23 (12.6) 159 (87.4)	0.065
Lymph node metastasis Present Absent	39 (46.4) 45 (53.6)	107 (58.8) 75 (41.2)	0.060
Distant metastasis Present Absent	9 (10.7) 75 (89.3)	20 (11.0) 162 (89.0)	0.947
pTNM stage I–II III–IV	42 (50.0) 42 (50.0)	73 (40.1) 109 (59.9)	0.130

Numbers in parentheses represent percentage.

**Table 2 medicina-57-00938-t002:** The correlation between REG4 expression and tumor stroma in colorectal cancers.

	REG4 Expression		*p*-Value
	Positive	Negative	
Stroma High Low	16 (19.0) 68 (81.0)	65 (35.7) 117 (64.3)	0.006
Mucinous component Present Absent	32 (38.1) 52 (61.9)	14 (7.7) 168 (92.3)	<0.001

Numbers in parentheses represent percentage.

## Data Availability

No new data were created or analyzed in this study. Data sharing is not applicable to this article.

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
