# Peer review of "Clinicopathological Significance and Prognostic Implications of REG4 Immunohistochemical Expression in Colorectal Cancer"

_medicina, 2021, doi:10.3390/medicina57090938_

Round 1
Reviewer 1 Report
In this study, Guhyun Kang and colleagues investigated the effect of REG4 immunohistochemical expression in colorectal cancer (CRC) on clinicopathological significance and prognosis. In this study, 266 CRC tissues undergoes immunohistochemical analysis and the results showed REG4 expression was significantly correlated with tumor location, amount of stroma and mucinous component in CRC. Few suggestions for this study are:
- Immunohistochemistry analysis and qualitative categorization of the samples according to staining and prognosis of colorectal cancer should be done in blinded fashion. Please mentioned in the limitation.
- It is interesting to see in low stroma group has significantly higher REG4 expression than high stroma group, but in high stroma group has REG4 expression significantly correlated with poor survival while low stroma group shows no significant difference. How to explain this result? Mechanism of REG4?
- Organization of Introduction: Correlation between HIFα and REG4 is presented in results and discussion, it should be mentioned in introduction.
- Claim of being the first to elucidate the clinocopathological and prognostic implications of REG4 expression through immunohistochemical analysis of human CRC tissues? Oue et al study has examined the expression and distribution of Reg IV in human CRC by immunohistochemistry, and the relationship between Reg IV staining and clinicopathological characteristics.
Author Response
In this study, Guhyun Kang and colleagues investigated the effect of REG4 immunohistochemical expression in colorectal cancer (CRC) on clinicopathological significance and prognosis. In this study, 266 CRC tissues undergoes immunohistochemical analysis and the results showed REG4 expression was significantly correlated with tumor location, amount of stroma and mucinous component in CRC. Few suggestions for this study are:
Immunohistochemistry analysis and qualitative categorization of the samples according to staining and prognosis of colorectal cancer should be done in blinded fashion. Please mentioned in the limitation.
Response:
As the reviewer’s recommendation, we added the comment for the limitation in the revised manuscript as below:
The immunohistochemistry results were evaluated by two independent pathologists (G.Kang and J.S.Pyo). Cases with inconsistent result were reviewed and confirmed by agreement between two pathologists. However, for the qualitative categorization of the samples, the interpretation of blinded fashion could not be performed.
It is interesting to see in low stroma group has significantly higher REG4 expression than high stroma group, but in high stroma group has REG4 expression significantly correlated with poor survival while low stroma group shows no significant difference. How to explain this result? Mechanism of REG4?
Response:
We described in the discussion as below:
Although CRC with high stroma was significantly correlated with low REG4 expression, patients with CRCs with high stroma and REG4 expression showed worse survival than other CRC patients. REG4 expression could be stratified according to the prognosis of CRCs with high stroma. The prognostic implication of REG4 expression can differ according to the amount of intratumoral stroma. Therefore, in a study regarding the clinicopathological significance of REG4 expression, it is important to evaluate the amount of stroma. Thus, the interaction between REG4 expression and the stroma may be important.
To elucidate the correlation between the amount of stroma and REG4 expression, we performed the additional investigation. We showed the results and descriptions as below:
We also studied the correlation between REG4 and HIF-1α expression in CRC. HIF-1α expression was significantly higher in CRCs with high stroma and REG4 expression than that in the other subgroups. However, in the present study, there were no significant differences in survival between patients with CRCs with and without HIF-1α expression. The correlation between REG4 and HIF-1α expression has not been previously reported. Although our study could not completely elucidate the detailed pathway, it is the first to demonstrate the correlation between REG4 and HIF-1α expression through immunohistochemical analysis.
In the present study, we evaluated the prognostic roles of REG4 immunohistochemical expression according to the subgroup (the amount of stroma). Each result, (1) the correlation between the amount of stroma and REG4 expression; (2) the correlation between the prognosis and REG4 expression in stroma-low subgroup; and (3) the correlation between the prognosis and REG4 expression in stroma-high subgroup, was independent and cannot be interpreted as conflicting results.
Organization of Introduction: Correlation between HIFα and REG4 is presented in results and discussion, it should be mentioned in introduction.
Response:
As the recommendation, we mentioned in the revised manuscript as below:
In addition, to elucidate the correlation between REG4 and stroma of tumor, the implica-tion of HIF-1α expression was investigated.
Claim of being the first to elucidate the clinocopathological and prognostic implications of REG4 expression through immunohistochemical analysis of human CRC tissues? Oue et al study has examined the expression and distribution of Reg IV in human CRC by immunohistochemistry, and the relationship between Reg IV staining and clinicopathological characteristics.
Response:
As the recommendation, we corrected the mention as below:
The present study is to elucidate the clinicopatho-logical and prognostic implications of REG4 expression through immunohistochemical analysis of human CRC tissues. In addition, the correlation between REG4 expression and tumor stroma was investigated.
Reviewer 2 Report
Kang et al. present an immunohistochemical study on REG4 in colorectal carcinomas. In total 266 cases were included. The study design is well presented. IHC experssion is correlated with topgraphy, other morphological features and prognosis. The results are well presented with tables, graphs and microphographs. They showed correlation with right sided tumors, low stroma and mucinous component. Correlation was also found with prognosis.
The results are discussed in the light of the recent literature. I would like to see more profound discussion on inflammation, especially tumor associated and chemo- and radiotherapy treatment.
Author Response
Kang et al. present an immunohistochemical study on REG4 in colorectal carcinomas. In total 266 cases were included. The study design is well presented. IHC experssion is correlated with topgraphy, other morphological features and prognosis. The results are well presented with tables, graphs and microphographs. They showed correlation with right sided tumors, low stroma and mucinous component. Correlation was also found with prognosis.
The results are discussed in the light of the recent literature. I would like to see more profound discussion on inflammation, especially tumor associated and chemo- and radiotherapy treatment.
Response:
As the recommendation, we performed the additional analysis for the correlation between REG4 expression and tumor-infiltrating lymphocytes (TILs). However, there was no significant correlation between REG4 expression and TILs (as below Supplementary Table).
|
Supplementary Table |
|||
|
|
REG4 expression |
P-value |
|
|
Positive |
Negative |
||
|
Immunoscore High Low |
33 (39.3) 51 (60.7) |
75 (41.2) 107 (58.8) |
0.790 |
|
CD3-positive lymphocytes High Low |
30 (35.7) 54 (64.3) |
63 (34.6) 119 (65.4) |
0.890 |
|
CD8-positive lymphocytes High Low |
28 (33.3) 56 (66.7) |
66 (36.3) 116 (63.7) |
0.681 |
|
PD-L1 expression of tumor cells ≥10% <10% |
5 (6.0) 79 (94.0) |
20 (11.0) 162 (89.0) |
0.259 |
|
PD-L1 expression of immune cells ≥10% <10% |
11 (13.1) 73 (86.9) |
36 (19.9) 145 (80.1) |
0.227 |
|
Numbers in parentheses represent percentage |
|||
In cancer treatment, immunotherapy is an important treatment in recent years. So, we analyzed the correlation between REG4 expression and PD-L1 expression of tumor and immune cells. In addition, there was no significant correlation between REG4 expression and PD-L1 expression. Based on our supplementary results, after the malignant transformation, the impact of REG4 on the inflammatory process can be limited. In addition, the roles of REG4 is not clear in chemo- and radiotherapy of other malignant tumors
We added the additional findings and comments in the revised manuscript as below:
Moreover, because REG4 can be correlated with inflammation, the correlation between REG4 and tumor-infiltrating lymphocytes was investigated. REG4 expression was not significantly correlated with tumor-infiltrating lymphocytes (P = 0.790; data not shown). After the malignant transformation, the impact of REG4 on the inflammatory process can be limited. In addition, REG4 expression was not correlated with PD-L1 expressions of tumor and immune cells (P = 0.259 and P = 0.227, respectively; data not shown).

Round 2
Reviewer 1 Report
The authors revised the MS accordingly.
Author Response
The authors revised the MS accordingly.
Response: Thank you for the careful review.